# Functional Characterization of ABCA4 Missense Variants Linked to Stargardt Macular Degeneration

**DOI:** 10.3390/ijms22010185

**Published:** 2020-12-27

**Authors:** Fabian A. Garces, Jessica F. Scortecci, Robert S. Molday

**Affiliations:** 1Department of Biochemistry and Molecular Biology, University of British Columbia, Vancouver, BC V6T 1Z3, Canada; fabiangarces111@gmail.com (F.A.G.); jessy.fernandes19@gmail.com (J.F.S.); 2Department of Ophthalmology and Visual Sciences, University of British Columbia, Vancouver, BC V5Z 3N9, Canada

**Keywords:** ABC transporter, ABCA4, Stargardt macular degeneration, ATPase activity, photoreceptors, inherited retinal diseases, cell expression, molecular modeling

## Abstract

ABCA4 is an ATP-binding cassette (ABC) transporter expressed in photoreceptors, where it transports its substrate, *N*-retinylidene-phosphatidylethanolamine (*N*-Ret-PE), across outer segment membranes to facilitate the clearance of retinal from photoreceptors. Mutations in *ABCA4* cause Stargardt macular degeneration (STGD1), an autosomal recessive disorder characterized by a loss of central vision and the accumulation of bisretinoid compounds. The purpose of this study was to determine the molecular properties of ABCA4 variants harboring disease-causing missense mutations in the transmembrane domains. Thirty-eight variants expressed in culture cells were analyzed for expression, ATPase activities, and substrate binding. On the basis of these properties, the variants were divided into three classes: Class 1 (severe variants) exhibited significantly reduced ABCA4 expression and basal ATPase activity that was not stimulated by its substrate *N*-Ret-PE; Class 2 (moderate variants) showed a partial reduction in expression and basal ATPase activity that was modestly stimulated by *N*-Ret-PE; and Class 3 (mild variants) displayed expression and functional properties comparable to normal ABCA4. The p.R653C variant displayed normal expression and basal ATPase activity, but lacked substrate binding and ATPase activation, suggesting that arginine 653 contributes to *N*-Ret-PE binding. Our classification provides a basis for better understanding genotype–phenotype correlations and evaluating therapeutic treatments for STGD1.

## 1. Introduction

ABCA4 is a member of the A-subfamily of ATP-binding cassette (ABC) transporters primarily expressed in rod and cone photoreceptors [1,2,3]. Like other members of this subfamily, ABCA4 is organized as two nonidentical tandem halves, with each half consisting of a transmembrane domain (TMD), a nucleotide binding domain (NBD), and large exocytoplasmic domains (ECD) [4]. ABCA4 functions as a lipid importer, flipping N-retinylidene-phosphatidylethanolamine (*N*-Ret-PE), the Schiff base adduct of retinal and phosphatidylethanolamine (PE), from the lumen to the cytoplasmic leaflet of photoreceptor outer segment disc membranes [5,6]. This transport activity ensures that all-trans retinal (ATR) generated from photoexcitation and 11-cis retinal not needed for the regeneration of rhodopsin and cone opsins are effectively cleared from the photoreceptors by the visual cycle.

The importance of ABCA4 is highlighted by the finding that mutations in the gene encoding ABCA4 are responsible for autosomal recessive Stargardt disease (STGD1:MIM 248200) as well as some recessive forms of cone-rod dystrophy and retinitis pigmentosa [1,7,8,9,10,11,12]. Stargardt disease, the most common inherited macular degeneration, is characterized by loss in central vision, progressive bilateral atrophy of the macula including the underlying retinal pigment epithelial (RPE) cells, impaired color vision, delayed dark adaptation, and the accumulation of fluorescent yellow-white flecks around the macula and mid-retinal periphery. Disease severity varies widely from severe loss in central vision in the first decade of life to mild visual impairment late in life. Over 1000 mutations in ABCA4 are now known to cause ABCA4-related diseases (https://databases.lovd.nl/shared/genes/ABCA4). These include missense mutations, frameshifts, truncations, small deletions, insertions, splice site mutations, and deep intronic mutations. Over 60% are missense mutations, resulting in amino acid substitutions spread throughout the protein [7,8].

A number of disease-associated variants with amino acid substitutions in the exocytoplasmic domains (ECDs), the nucleotide-binding domains (NBDs), and the C-terminal segment of ABCA4 have been previously characterized [13,14,15,16,17,18,19]. Many of these variants result in a significant reduction in cellular expression, mislocalization of ABCA4, and a loss of functional properties including substrate-stimulated ATPase activity and ATP-dependent substrate transport activity. Loss of ABCA4 transport activity results in the formation and accumulation of bisretinoids, including A2E in the photoreceptors and RPE cells [20]. These compounds accumulate as fluorescent lipofuscin deposits in STGD1 patients and animal models harboring ABCA4 null alleles and loss of function missense mutations [17,21,22,23,24,25,26]. To date, however, the effect of disease-causing missense variants in the TMDs on the functional properties of ABCA4 has not been examined in detail.

In this study, we have analyzed the biochemical properties of 38 disease-causing variants in TMD1 and TMD2 of ABCA4 including the membrane spanning segments, a proposed coupling helix, and exocytoplasmic V-shaped hairpin helices. We show that a significant fraction of these variants exhibits decreased ABCA4 expression due to protein misfolding, leading to a significant reduction in *N*-Ret-PE binding and ATPase activity. In contrast, some mutations only marginally affect the expression and functional properties of ABCA4. The loss of expression and activity can be used to predict the severity of STGD1 in individuals harboring these disease-associated mutations. As part of this study, we have also identified an arginine at position 653 (R653) in transmembrane segment 2 (TM2) as a residue that directly contributes to the binding of *N*-Ret-PE to ABCA4.

## 2. Results

### 2.1. Transmembrane Domains (TMDs) of ABCA4

Although earlier sequence analysis coupled with biochemical studies defined the topological organization of ABCA transporters within membranes, the exact location of the membrane spanning segments was not well-defined in this model [4]. We have now pooled together several programs that predict transmembrane segments including DAS-TMfilter, ExPASy TMpred, HMMTOP, MP Toppred, PredictProtein, and TMHMM (https://www.expasy.org/tools/) in an effort to more definitively identify the membrane spanning segments of ABCA4 (Figure 1A). These assignments were further supported through analysis of the transmembrane segments defined in the cryo-electron structure of ABCA1 [27]. This member of the ABCA subfamily of ABC transporters shares about 50% sequence identity with ABCA4. This high degree of identity extends to TMDs, with TMD1 sharing a 51% identity and TMD2 sharing a 60% identity (Figure 1B). On the basis of this high degree of sequence identity, we have generated a homology model of ABCA4 from the structure of ABCA1 (Figure 1C). Not surprisingly, the ABCA4 model predicts the location of intracellular transverse coupling helices (IH) and exoplasmic V-shaped α-helical hairpin helices (EH) as initially identified in ABCA1 [27]. The location of the disease-associated missense variants in the TMDs of ABCA4 analyzed in the present study is shown in Figure 1A,B,D.

### 2.2. Expression of TMD Disease Variants

The expression of 38 ABCA4 disease-associated variants in transfected HEK29 cells was quantified by Western blotting after solubilization in the strong detergent SDS as a measure of total expression and after solubilization in the mild detergent CHAPS as a measure of more native-like protein. All TMD1 and TMD2 missense variants, expressed at similar levels when the cells solubilized with SDS were analyzed on Western blots labeled for ABCA4 (Figure A1). However, significant differences in protein levels were observed for cells solubilized in CHAPS and subjected to high-speed centrifugation to remove poorly solubilized, aggregated protein prior to Western blotting (Figure 2). Ten variants in TMD1 (p.L661R, p.L686S, p.G690V, p.S765N, p.S765R, p.V767D, p.L797P, p.M840R, p.D846H, p.G851D) solubilized below 40% WT ABCA4 levels; three (p.G818E, p.W821R, p.I824T) in the range of 40% to 52% WT ABCA4; and the remaining six (p.R653C, p.R653H, p.T716M, p.C764Y, p.V849A, p.A854T) at levels above 80% WT ABCA4 (Figure 2A, Table 1). In the case of TMD2, three variants solubilized at levels below 40% of WT ABCA4 (p.A1773E, p.A1773V, p.H1838R); seven variants between 40% and 80% WT (p.P1380L, p.Q1703K, p.R1705L, p.A1794D, p.A1794P, p.H1838D, p.H1838Y); and nine variants (p.E1399K, p.S1696N, p.Q1703E, p.H1838N, p.N1805D, p.R1843W, p.N1868I, p.R1898C, p.R1898H) above 80% WT (Figure 2B, Table 2). These results suggest that many TMD variants result in highly misfolded and aggregated proteins that fail to solubilize in CHAPS detergent, resulting in the decreased levels of ABCA4 expression observed by Western blotting.

### 2.3. Localization of ABCA Disease Variants in COS-7 Cells

The distribution of ABCA4 in transfected cells was examined by immunofluorescence microscopy. For these experiments, we used COS-7 cells as these cells have a large, flat surface area, making them a preferable cell line for microscopy. Previously, we showed that ABCA4 variants with low solubilization levels in CHAPS tend to be retained in the endoplasmic reticulum (ER) of transfected cells, whereas variants that express at near WT levels accumulate in intracellular vesicle-like structures [14,16]. Of the 38 variants examined, 13 variants in TMD1 and 10 variants in TMD2, which solubilized in CHAPS at levels below 80% WT ABCA4, showed a reticular distribution in COS7 cells and co-localized with the ER marker calnexin, while TMD variants that solubilized above 80% WT levels displayed vesicular localization with some reticular labeling (Figure 3A–C). These studies support the view that a significant fraction of the TMD variants that exhibit low solubilization in a mild detergent are misfolded and retained in the ER by the quality control system of the cell, while TMD variants showing a high degree of solubilization by CHAPS can exit the ER and accumulate in vesicle-like structures similar to WT ABCA4.

### 2.4. Functional Analysis of TMD Disease-Associated Variants: ATPase Assays

The ATPase activities and substrate binding properties of the ABCA4 variants were measured in order to determine the effect of TMD disease-associated mutations on the functional properties of ABCA4. For the ATPase assays, the ABCA4 variants expressed in HEK293T cells and solubilized in CHAPS detergent were purified on an immunoaffinity matrix. An aliquot of each variant was analyzed by SDS gel electrophoresis to ensure that all variants had a similar degree of purity and concentration for use in the ATPase assays. All ABCA4 variants migrated as the major Coomassie blue stained band with an apparent molecular mass of 250 kDa, supporting the view that all variants could be efficiently purified by immunoaffinity chromatography with a purity of over 90% (Figure 4A).

The ATPase activity of detergent-solubilized WT ABCA4 and disease-associated variants was measured in the absence and presence of all-trans retinal (ATR). In the absence of ATR, but in the presence of phosphatidylethanolamine (PE), WT ABCA4 displays basal activity which largely reflects the energy-dependent conformational change related to the flipping of the secondary substrate PE across membranes [15]. Addition of 40 µM ATR in the presence of PE leads to the formation of the primary substrate *N*-Ret-PE and a two-fold increase in ATPase activity in agreement with previous reports [13,14,19,28,29] (Figure 4B,C).

The ATPase activity of the TMD variants can be divided into three distinct groups. Group 1 includes variants that have basal ATPase activity below 50% of WT levels and show little or no *N*-Ret-PE-induced stimulation in ATPase activity; Group 2 comprises variants with basal ATPase activity between 50% and 80% of WT levels and shows modest *N*-Ret-PE-stimulated ATPase activity; and Group 3 consists of variants that have both basal and *N*-Ret-PE-stimulated ATPase activity comparable to WT. For TMD1, 10 variants (p.L661R, p.L686S, p.G690V, p.S765N, p.S765R, p.V767D, p.L797P, p.M840R, p.D846H, p.G851D) are classified in Group 1; 4 variants (p.G818E, p.W821R, p.I824T, p.A854T) belong to Group 2; and 4 variants (p.R653H, p.T716M, p.C764Y, p.V849A) are in Group 3 (Figure 4B, Table 1). For TMD2, 7 variants (p.Q1703E, p.Q1703K, p.A1773E, p.A1773V, p.A1794D, p.H1838R, p.H1838Y) belong to Group 1; 7 variants (p.P1380L, p.R1705L, p.A1794P, p.N1805D, p.H1838D, p.H1838N, p.R1843W) to Group 2; and 5 variants (p.E1399K, p.S1696N, p.N1868I, p.R1898C, p.R1898H) to Group 3 (Figure 4C, Table 2). The ATPase activities typically agreed with the degree of solubilization with some exceptions. In particular, the p.R653C variant expresses and displays basal ATPase activity similar to WT, but shows no significant substrate-activated ATPase activity (Figure 4B; also see Figure 8).

To further define how various disease-associated genetic mutations affect the ATPase activity of ABCA4, we measured the specific ATPase activity as a function of ATR concentration for selected Group 3 variants. For TMD1, p.T716M and p.V849A variants showed specific ATPase activity profiles similar to WT ABCA4, while p.C764Y variants had higher activity (Figure 5A). The p.A854T variant in Group 2, which solubilizes in CHAPS at a WT level, had a lower specific ATPase activity than WT (Figure 5A). For the Group 3 variants in TMD2, two variants (p.N1868I, p.R1898C) had similar specific ATPase activities to WT, while the p.S1696N variant had higher specific activity and the p.R1843W had a lower ATPase activity (Figure 5B).

### 2.5. Functional Analysis of Disease-Associated TMD Variants: N-Ret-PE Binding Assays

The effect of TMD mutations on the binding of *N*-Ret-PE to ABCAA4 was investigated using a solid phase binding protocol [14,16]. ABCA4 immobilized on an immunoaffinity matrix was treated with ATR in the presence of PE to generate *N*-Ret-PE. After the removal of excess substrate, the matrix was incubated in the absence or presence of 1 mM ATP and the amount of bound *N*-Ret-PE was determined. In the absence of ATP, *N*-Ret-PE tightly bound to WT ABCA4 as previously reported [10]. The addition of ATP induced a conformational change in ABCA4, resulting in an efficient release of over 85% of the bound substrate.

The *N*-Ret-PE binding profile for the TMD variants are shown in Figure 6A,B. Generally, the TMD variants could be separated into three main groups: One group bound *N*-Ret-PE at or below 40% WT levels, a second group bound *N*-Ret-PE in the range of 40–70% WT levels, and a third group bound *N*-Ret-PE above 70% WT levels. For TMD1, all 19 variants showed various levels of diminished *N*-Ret-PE binding. Fifteen variants (p.R653C, p.R653H, p.L661R, p.L686S, p.G690V, p.S765N, p.S765R, p.V767D, p.L797P, p.G818E, p.W821R, p.I824T, p.M840R, p.D846H, p.G851D) displayed very low substrate binding and little or no substrate release upon binding with ATP. These included the variants that had low basal and *N*-Ret-PE-stimulated ATPase activity. Four variants (p.T716M, p.C764Y, p.V849A, p.A854T) bound *N*-Ret-PE in the range of 40–70% WT levels. For TMD2, however, the opposite was true, with 15 of the 19 variants showing substrate binding above 40% WT ABCA4. Five of these variants (p.E1399K, p.S1696N, p.R1843W, p.N1868I, p.R1898H) displayed *N*-Ret-PE-binding and release similar or, in some cases, greater than WT ABCA4 (Figure 6B, Table 2).

Substrate binding curves were constructed to further investigate the apparent affinity of selected variants for *N*-Ret-PE. For the TMD1 variants p.T716M, p.C764Y, p.V849A, and p.A854T, the apparent dissociation constant Kd was 4 to 7 times higher than for WT ABCA4, indicating that these variants had significantly lower affinity for its substrate than WT ABCA4 (Figure 7A). For the TMD2 variants, there was a wide range in Kd values. The apparent Kd for the p.S1696N variant was 3 times higher than for WT ABCA4, while the Kd for the p.N1868I, p.R1898C, and p.R1843W variants were 12, 14, and 26 times higher, respectively (Figure 7B). These results indicate that the affinity of the TMD variants for *N*-Ret-PE varies widely, with many showing only a modest decrease in affinity of 3- to 5-fold (Kd 2.8–6.3 µM) and others showing a significantly larger decrease in affinity (Kd 8.6–31.1 µM). These results also emphasize that both TMDs contribute to the efficient binding and release of *N*-Ret-PE, although amino acid substitutions in TMD1 appear to affect substrate binding more than substitutions in TMD2.

### 2.6. Biochemical Characterization of ABCA4 Arg653 Variants

The p.R653C and p.R653H disease-linked variants displayed distinct biochemical properties when compared to other TMD variants. Both variants expressed and solubilized in mild detergent at WT levels (Figure 8A), displayed a WT-like vesicular expression pattern in transfected COS-7 cells (Figure 3C), and displayed a basal ATPase activity similar to WT ABCA4 (Figure 8B). However, the basal ATPase activity of the p.R653C variant was not statistically activated by *N*-Ret-PE (*p* > 0.05) as shown in Figure 8B, and this variant was deficient in *N*-Ret-PE binding (Figure 8C). In contrast, the ATPase activity of the p.R653H variant was stimulated by *N*-Ret-PE, and this variant displayed significant substrate binding in the absence of ATP, although at a reduced level relative to WT ABCA4 (Figure 8C). These properties prompted us to further explore the effect of other amino acid substitutions at position 653 of ABCA4, with the objective to determine if a positively charged residue at this position is required for substrate binding and ATPase activation.

For these studies, three additional variants containing a neutral (p.R653A), a negatively charged (p.R653E), and a positively charged (p.R653K) residue were constructed and their properties were compared to the p.R653C and p.R653H variants. All these variants solubilized at or close to WT levels and formed vesicle-like structures in transfected COS-7 cells, indicative of proper protein folding (Figure 3C, Figure 4B and Figure 8A). They also retained basal ATPase activity levels similar to WT ABCA4 levels (Figure 8B). However, only the variants with positively charged residues (p.R653H and p.R653K) displayed statistically significant *N*-Ret-PE-induced increase in ATPase activity, with the p.R653H variant showing statistically higher basal and substrate-stimulated activity than even WT ABCA4 (Figure 8B). The *N*-Ret-PE binding properties of p.R653C, p.R653A, and p.R653E were drastically decreased, but only moderately reduced for p.R653K (Figure 8C). The substrate binding affinity of these variants was also measured. The apparent Kd of p.R653K was only 2-fold greater than WT ABCA4 (Kd 2.5 µM for p.R653K compared to 1.2 µM for WT ABCA4), as shown in Figure 8C. However, the p.R653H variant had a significantly lower affinity with a Kd of 13.3 µM. Taken together, these results indicate that a positively charged residue at position 653 is important for significant and stable *N*-Ret-PE binding and ATPase activation.

## 3. Discussion

In this study, we analyzed 38 disease-causing missense variants with amino acid substitutions in the two TMDs in order to evaluate the effect of these changes on the expression and functional properties of ABCA4 and define pathogenic mechanisms responsible for STGD1. The majority of the genetic mutations caused amino acid replacements within the membrane, spanning segments of TMD1 and TMD2 (Figure 1). Two mutations, however, were present in the cytoplasmic loop containing the intracellular transverse helix IH2 connecting TM2 and TM3 and six mutations were situated in the extracellular membrane-penetrating V-shaped helices EH1-EH2 in TMD1 and EH3-EH4 in TMD2, based on our homology model generated from the cryo-EM structure of ABCA1 and amino acid alignment of these ABCA transporters within the TMDs (Figure 1A,B,D). Our results indicate that many of the disease-causing TMD mutations lead to protein misfolding, decreased substrate binding, and diminished basal and substrate-activated ATPase activity or a combination of these properties.

The residual activity of disease-associated ABCA4 variants and their level of expression are important factors in understanding protein structure–function relationships, pathogenic mechanisms, and genotype–phenotype correlations. The function of the ABCA4 variants is best determined by measuring the ATP-dependent flipping of *N*-Ret-PE across membranes, an activity required for the removal of toxic retinal compounds from photoreceptors [6,30]. However, *N*-Ret-PE transport assays are difficult to carry out, particularly when there is a need to evaluate large numbers of variants. Instead, we have used the activation of the basal ATPase activity by *N*-Ret-PE as a measure of ABCA4 function, since this assay is straightforward to carry out, sensitive, and mirrors the ATP-dependent substrate transport activity of ABCA4 [6,13,31]. The degree to which a mild detergent solubilizes ABCA4 variants from the membranes of transfected cells was used as a measure of the expression levels of the variants. The expression level of one disease-associated variant, N965S, in culture cells has been shown to agree with the expression levels of ABCA4 in mice homozygous for this variant [22], supporting the use of culture cells to measure ABCA4 variant expression.

On the basis of ATPase activity measurements, substrate binding properties, and expression levels, the TMD variants can be separated into three major classes (Table 1 and Table 2). Class 1 consists of ABCA4 variants that typically show highly diminished expression and basal ATPase activities below 50% of the WT ABCA4 level with little if any *N*-Ret-PE stimulation of ATPase activity or *N*-Ret-PE binding and release by ATP. These loss-of-function variants are predicted to confer an early disease onset, typically within the first decade of life, and a severe progressive visual impairment in individuals homozygous for these variants or compound heterozygous with another severe loss-of-function variant. Class 2 includes variants that have reduced expression and basal ATPase activities in the range of 50–80% WT ABCA4, but show modest *N*-Ret-PE binding and stimulation of basal ATPase activity. These variants are predicted to convey a more moderate disease phenotype with an age of onset in the second to fifth decade of life. Class 3 consists of variants with significant expression levels and basal ATPase activity typically greater than 80% WT levels and robust *N*-Ret-PE-stimulated ATPase activity. These variants are predicted to impart a late disease onset and mild visual impairment. When paired with another mild variant, these Class 3 variants may or may not confer a disease phenotype depending on the combined activity of these variants. In some instances, these variants may be considered as hypomorphic variants.

Of the 38 TMD variants studied here, 18 ABCA4 variants (11 in TMD1 and 7 in TMD2) comprise Class 1 (Table 1 and Table 2). In almost all cases, the low basal and substrate-stimulated ATPase activity of these variants coincides with a loss in ATP-dependent *N*-Ret-PE binding and a low level of expression after mild detergent solubilization. These properties are consistent with these amino acid changes causing extensive protein misfolding. This is further supported by the finding that these variants are retained in the ER of transfected cells by the quality control system of the cell, as visualized by immunofluorescence microscopy. Indeed, most of these missense mutations involve an amino acid substitution which alters the charge or polarity of the side chain. Such a change can adversely affect the folding and packing of the transmembrane helices, the interface between the side chains and the hydrophobic lipid bilayer, or the conformation of loops connecting the membrane spanning segments. Interestingly, two disease-associated mutations (p.L686S and p.G690V) in Class 1 reside in the cytoplasmic loop between TM2 and TM3, containing a proposed intracellular coupling helix (IH2) which is expected to coordinate ATP binding and hydrolysis in the NBDs with the transport of substrate through the TMDs (Figure 9A). Glycine at position 690 is conserved in ABCA4 from other vertebrates as well as ABCA1 and ABCA7 (Figure 1B). Leucine at position 686 has a nonpolar methionine at this position in ABCA1 and ABCA7 as well as in some ABCA4 orthologues, supporting the importance of an amino acid with a hydrophobic side chain within this structural motif. One mutation (p.L797P) in Class 1 variants and three (p.G818E, p.I824T, p.W821R) in Class 2 variants are situated in highly conserved regions within the exocytoplasmic loop between TM5 and TM6, and another five (p.H1838D/N/R/Y and p.R1843W) in Group 1 and 2 reside in a conserved region within the exocytoplasmic loop between TM11 and TM12. These two loops contain the V-shaped α-helical hairpin, as initially reported for the structure of ABCA1 [27] and found in the homology model of ABCA4 reported here (Figure 9B,C). Residues L797, G818, W821, H1838, and R1843 in ABCA4 are also present in ABCA1 and ABCA7 and are part of highly conserved motifs supporting the importance of these structural features in the proper folding and function of these transporters (Figure 9B,C). In the case of residue H1838, four variants (p.H1838D, p.H1838N, p.H1838R, p.H1838Y) have been reported to cause STGD1 and are shown here to have moderate to severe effects on the functional properties of ABCA4, suggesting that the histidine at position 1838 plays a crucial role in maintaining the optimal structure and function of ABCA4.

Our classification, based on functional data, can be used to predict the impact that missense mutations have on the severity of STGD1. Unfortunately, sufficient metadata for large cohorts of patients including age of onset, visual acuity, autofluorescence, or other diagnostic properties have not been reported for most of the TMD mutations examined here to make reliable correlations between the genotype and residual functional activity of the variants. However, such data are available in a few cases such as, for example, the p.P1380L Class 2 variant within TM7 (Figure 1). The age of onset for individuals homozygous for this variant ranges from 10 to 26 years, with an average age of 19 years [9,32,33,34]. This age of onset is consistent with the moderate disease classification presented here and the moderate classification based on clinical findings [35]. When this mutation is in trans with the p.S1696N mutation, reported here to be a mild Class 3 mutation, then the age of onset is increased to 45 years, supporting our Class 3 designation of the p.S1696N variant [33].

The p.N1868I mild variant is another for which there is considerable genetic and clinical data [36,37]. The genetic frequency and penetrance of the p.N1868I mutations have led to considerable discussion about whether this variant is benign or pathogenic [7]. Support for the pathogenic nature of this variant comes from the analysis of STGD1 patients harboring the p.N1868I variant in trans with a deleterious mutation. The age of onset varies significantly between 18 and 72, even in patients with the same genotype (e.g., p.N1868I/p.L257Vfs*17) [36,37]. Our biochemical studies indicate that p.N1868I exhibits an expression and activity profile broadly similar to WT ABCA4. However, the affinity of this variant for *N*-Ret-PE is an order of magnitude lower than that of WT ABCA4, suggesting that this mutation alters the substrate binding properties of ABCA4. The relatively low affinity of this variant for *N*-Ret-PE would compromise the efficient transport of *N*-Ret-PE across disc membranes, leading to the formation of bisretinoids in photoreceptors and gradual accumulation in RPE cells, particularly when this variant is paired with a functionally deficient variant. The p.N1868I mutation is situated in a loop linking the V-shaped hairpin helices to TM12 in TMD2 (Figure 9C). Other mild variants include p.T716M, p.C764Y, p.V849A, p.E1399K, p.S1696N, and p.R1898H/C. In some cases, a similar amino acid substitution occurs such as, for example, p.V849A, p.S1696N, and p.R1898H, whereas in other cases, the amino acid substitutions may have little impact on the structure or function of ABCA4 based on their location within the structure, such as in the case of the p.E1399K and p.R1898C/H variants.

Of the variants examined in this study, the p.R653C Class 1 variant is particularly interesting in that it shows unique properties. This variant solubilized in CHAPS buffer and localized within vesicle-like structures in transfected cells similar to WT ABCA4. Furthermore, the basal ATPase activity is similar to WT ABCA4. However, the *N*-Ret-PE binding and stimulation of ATPase activity of this variant are severely impaired. These results strongly suggest that this mutation does not affect protein folding, but instead alters the interaction of *N*-Ret-PE with ABCA4. Additional substitutions at this position further highlight the requirement for an amino acid with a positively charged side chain at position 653 to facilitate *N*-Ret-PE binding. The p.R653A variant shows drastically reduced *N*-Ret-PE binding and ATPase activation and the p.R653E variant lacks these properties, whereas the p.R653K and p.R653H variants show significant substrate binding and ATPase activation. Although the p.R653H variant has significant substrate-stimulated ATPase activity, this mutation has been reported to cause STGD1 [8]. This may result from the lower affinity of this variant for *N*-Ret-PE (apparent Kd of 13.3 µM for p.R653H vs. 1.2 µM for WT ABCA4) as shown in our binding studies. Our data suggest that the p.R653C variant with limited function would cause a more severe phenotype than the p.R653H variant. This is supported by studies reporting that STGD1 patients compound heterozygous for the p.R653C variant and a frameshift mutation (p.T1537Nfs*18) or a premature stop mutation (p.R2030*) have an age of disease onset within the first decade of life (6–10-years-old) [38,39]. On the basis of these studies, we suggest that arginine at position 653 located in TM2 close to the exocytoplasmic side of the membrane may form part of the binding pocket for *N*-Ret-PE. Modeling studies suggest that this binding site is close to the interface with ECD1 and the helical segment (EH2) present within the exoplasmic V-shaped hairpin structure between TM 5 and 6 TM5 (Figure 9D). In this respect, it is interesting to note that in the structure of ABCA1, electron density that may represent the phospholipid substrate was observed within a shallow pocket enclosed by the intracellular regions of TM1/2/5 [27]. The presence of the possible lipid binding site of ABCA1 at the interface with the cytoplasmic side and the proposed lipid binding site of ABCA4 on the exocytoplasmic side of the membrane is consistent with functional studies showing that ABCA1 is a lipid exporter with lateral access to the lipid substrate toward the cytoplasmic side, and ABCA4 is a lipid importer with lateral access to the lipid substrate toward the exocytoplasmic side of the membrane [15]. Additional studies are needed to more fully delineate the contributions of various amino acid residues within ABCA4 to *N*-Ret-PE binding.

In conclusion, we present here expression and functional data for a large set of disease-associated mutations which result in amino acid changes within the transmembrane domains of ABCA4. From these data, we have developed a classification which can be used to correlate the residual activity of the missense mutations with the severity of the disease. The information related to the functional properties and molecular mechanisms underlying TMD variants should serve as a basis for the development of novel gene, drug, and cell-based therapies for STGD1 disease.

## 4. Materials and Methods

### 4.1. Prediction of Transmembrane Helices and Homology Model of ABCA4

The amino acid sequence of the transmembrane (TM) helices that form the TMDs of ABCA4 were predicted using algorithms based on hidden Markov models (DAS-TMfilter, ExPASy TMpred, HMMTOP, MP Toppred, PredictProtein, and TMHMM) (https://www.expasy.org/tools/). The membrane spanning segment predictions from each program were determined and the predictions common to all or most programs were used as the most probable TM sequences in the TMDs. To validate the quality of these predictions, we used TMD sequences reported for the cryo-electron microscopic structure of ABCA1 [27].

Homology models of ABCA4 were made with I-TASSER and SWISS-MODEL using the ABCA1 Cryo-EM structure (5XYJ) as the template, which bears 50% sequence identity with ABCA4 [27]. For modeling with I-TASSER, ABCA4 was split into two tandem halves and each half was modeled individually [40]. The two halves had a 9-amino acid region of overlap just after the NBD1 (GDRIAIIAQ), and the two models were aligned and fused at this region of overlap using PyMol to construct the full homology model of ABCA4. Both programs gave comparable models.

### 4.2. Cloning of ABCA4 Transmembrane Variants

The cDNA of human ABCA4 (NM_000350) containing a 1D4 tag (TETSQVAPA) at the C-terminus was cloned into the pCEP4 vector using the Nhe-I and Not-I restriction sites as previously described [16]. Missense mutations were generated by PCR-based site-directed mutagenesis. All DNA constructs were verified by Sanger sequencing.

### 4.3. Antibodies

The Rho 1D4 monoclonal antibody was generated in house [41] and is available commercially (Millipore-Sigma, Oakville, ON, USA, Cat #MAB5356). It is used in the form of hybridoma culture fluid for Western blotting and immunofluorescence microscopy. For immunoaffinity purification, the purified Rho 1D4 antibody was covalently coupled to CNBr-activated Sepharose 2B as previously described [16].

### 4.4. Heterologous Expression of ABCA4 Variants in HEK293T Cells

HEK293T cells grown at 80% confluency in 10 cm plates were transfected with 10 µg of pCEP4-ABCA4-1D4 using 1 mg/mL PolyJetTM (SignaGen, Rockville, MD, USA) at a 3:1 PolyJetTM to DNA ratio. After 6 to 8 h, the cells were placed in fresh media. At 48 h post transfection, the cells were harvested and centrifuged at 2800× *g* for 10 min. The pellet was resuspended in 200 µL of resuspension buffer (50 mM HEPES, 100 mM NaCl, 6 mM MgCl_2_, 10% glycerol, pH 7.4). A 40 µL aliquot of the resuspended pellet was solubilized at 4 °C for 40 min in 500 µL of either 3-[(3-cholamidopropyl) dimethylammonio]-1-propanesulfonate hydrate (CHAPS) solubilization buffer (20 mM CHAPS, 50 mM HEPES, 100 mM NaCl, 6 mM MgCl_2_, 1 mM dithiothreitol (DTT), 1× ProteaseArrest, 10% glycerol, 0.19 mg/mL brain-polar-lipid (BPL), and 0.033 mg/mL 1,2-dioleoyl-sn-glycero-3-phospho-L-ethanolamine (DOPE) (Avanti Polar Lipids, Alabaster, AL), pH 7.4) or SDS solubilization buffer (3% SDS, 50 mM HEPES, 100 mM NaCl, 6 mM MgCl_2_, 1 mM DTT, 1× ProteaseArrest, 10% glycerol, 0.19 mg/mL BPL, and 0.033 mg/mL DOPE, pH 7.4). The samples were then centrifuged at 100,000× *g* for 10 min in a TLA110.4 rotor using a Beckman Optima TL centrifuge. The supernatant was collected and the absorbance at 280 nm was measured to determine protein concentration. Approximately 7–8 µg of total protein per lane were resolved on an 8% SDS-polyacrylamide gel and transferred onto a polyvinylidene difluoride membrane for Western blotting. The blots were blocked in 1% milk for 1 h, labeled for 2 h with culture fluid containing the Rho1D4 mouse monoclonal antibody (1:100 dilution in phosphate-buffered saline (PBS)) and rabbit-anti-tubulin (1:1000 dilution in PBS), and used as a loading control. The blots were washed 3 times with PBS followed by incubation for 1 h with donkey anti-mouse IgG or donkey anti-rabbit IgG conjugated to IR dye 680 (1:20,000 dilution in PBS-0.5%Tween (PBS-T). The blots were washed 3 times with PBS-T and imaged on an Odyssey Li-Cor imager (Li-Cor, Lincoln, NE, USA). Protein expression levels were quantified from the intensity of the ABCA4 bands and normalized to the intensity of the bands from the tubulin loading control.

### 4.5. Immunofluorescence Microscopy of ABCA4 Variants in Transfected COS-7 Cells

COS-7 cells were seeded 24 h before transfection on six-well plates containing coverslips coated with poly-L-lysine to promote cell adhesion to coverslips. The cells were transfected with 1 µg of DNA and 3 µL of 1 mg/mL PolyJetTM for 6 to 8 h before replacing with fresh media. At 48 h post transfection, the cells were fixed with 4% paraformaldehyde in 0.1 M phosphate buffer (PB), pH 7.4, for 25 min and washed 3 times with PBS. The cells were then blocked with 10% goat serum in 0.2% Triton X-100 and PB for 30 min. Primary antibody labeling was carried out for 2 h in 2.5% goat serum, 0.1% Triton X-100, and PB using the Rho1D4 antibody against the 1D4-tag and the calnexin rabbit-polyclonal antibody as an endoplasmic reticulum (ER) marker. The coverslips were washed 3 times with PB, followed by secondary labeling using Alexa-488 goat-anti-mouse Ig (for ABCA4), Alexa-594 goat-antirabbit Ig (for calnexin), and were counterstained for nuclei with DAPI for 1 h. The coverslips were washed 3 times with PB to remove excess antibody and subsequently mounted onto microscope slides with Mowiol mounting medium and kept in the dark at 4 °C. The microscope slides were visualized under a Zeiss (Oberkochen, Germany) LSM700 confocal microscope using a 40× objective (aperture of 1.3). Images were analyzed using Zeiss Zen software and ImageJ.

### 4.6. ATPase Assay

Depending on the ABCA4 variant, 1 to 3 ten-centimeter plates of HEK293T at 80% confluency were transfected as described above. At 24 h post transfection, the cells were harvested and centrifuged at 2800× *g* for 10 min. Depending on the number of plates transfected, the pellet was resuspended in 1–2 mL of CHAPS solubilization buffer for 60 min at 4 °C followed by a 100,000× *g* centrifugation for 10 min as described above. The supernatant was incubated with 70 µL of packed Rho1D4–Sepharose affinity matrix for 60 min at 4 °C. The beads were washed twice with 500 µL of column buffer (10 mM CHAPS, 50 mM HEPES, 100 mM NaCl, 6 mM MgCl_2_, 1 mM DTT, 1X ProteaseArrest, 10% glycerol, 0.19 mg/mL BPL, and 0.033 mg/mL DOPE, pH 7.4), transferred to an Ultrafree-MC spin column, and washed another five times with 500 µL of column buffer. Bound ABCA4 was eluted from the Rho1D4 matrix twice with 75 µL of 0.5 mg/mL 1D4 peptide in column buffer at 18 °C. When necessary, the absorbance at 280 nm was taken to calculate the protein concentration and the samples were diluted with column buffer to have equal protein concentration across all variants tested. Thirty microliters of purified ABCA4 was loaded onto an 8% acrylamide gel along with BSA standards to calculate the amount of ABCA4 protein in each sample.

ATPase assays were carried out using the ADP-GLOTM Max Assay kit (Promega, Madison, WI) according to the manufacturer’s guidelines. For each ABCA4 variant tested, 15 µL aliquots of purified ABCA4 (~100 ng of protein per tube) were added to six microcentrifuge tubes. One microliter of 0.8 mM all-trans retinal in column buffer (or column buffer alone) was added to half of the samples (performed in triplicate) to obtain a final concentration of 0 or 40 µM all-trans retinal. Each tube was incubated for 15 min at room temperature in the dark to allow the all-trans retinal to react with the PE and form the Schiff base adduct *N*-Ret-PE. Subsequently, 4 µL of a 1 mM ATP solution (in column buffer) was added and the samples were incubated at 37 °C for 40 min. The final concentrations of all-trans retinal and ATP in each sample were 40 µM (or 0 µM) and 200 µM, respectively. For all ATPase assays, the ATPase-deficient mutant ABCA4-MM, in which the lysine residues in the Walker A motif of nucleotide binding domain 1 (NBD1) and nucleotide binding domain 2 (NBD2) were substituted for methionine, was used to subtract non-specific ATPase activity. Typically, the MM values were less than 10% of the basal ATPase activity.

### 4.7. N-Ret-PE Binding Assay

Tritiated all-trans retinal was prepared by the method of Garwin and Saari [42,43]. [^3^H] all-trans retinal was mixed with unlabeled all-trans retinal to obtain a final concentration of 1 mM and a specific activity of 500 dpm/pmol. For a typical binding assay, depending on the ABCA4 variant, 2–5 ten-centimeter plates of transfected HEK293T cells were harvested 48 h post transfection and centrifuged for 10 min at 2800× *g* as described above. The pellet was resuspended and solubilized in 3 mL of CHAPS solubilization buffer for 40 to 60 min at 4 °C and subsequently centrifuged at 100,000× *g* for 10 min to remove unsolublized material. The supernatant was collected and divided in half. Each half was incubated with 80 µL of packed Rho1D4–Sepharose affinity matrix equilibrated in column buffer and mixed by rotation for 60 min at 4 °C. The affinity matrix was washed twice with 500 µL of column buffer and mixed with 250 µL of 10 µM [^3^H] all-trans retinal (500 dpm/pmol) in column buffer for 30 min at 4 °C. The matrix was then washed 6 times with 500 µL of column buffer. One sample was incubated with 1 mM ATP and the other was incubated in the absence of ATP for 15 min at 4 °C. The affinity matrices were washed 5 times with 500 µL of column buffer and transferred to an Ultrafree-MC (0.45 µm filter) spin column followed by another 5 washes of 500 µL of column buffer. Bound [^3^H] all-trans retinal was extracted with 500 µL of ice-cold ethanol with shaking at 500 rpm for 20 min at room temperature and counted in a liquid scintillation counter. Bound ABCA4 was eluted from the Rho1D4–bead matrix with 3% SDS in column buffer and resolved in an 8% SDS-polyacrylamide gel for analysis of protein levels by Western blotting. All washes and incubations with [^3^H] all-trans retinal were performed in the dark. In some studies, the concentration of all-trans retinal was varied between 0 and 50 µM.

### 4.8. Statistical Analysis

Statistical analysis of the data was carried out using GraphPad Prism 8.0. Data were expressed as a mean ± standard deviation (SD).

## Figures and Tables

**Figure 1 ijms-22-00185-f001:**
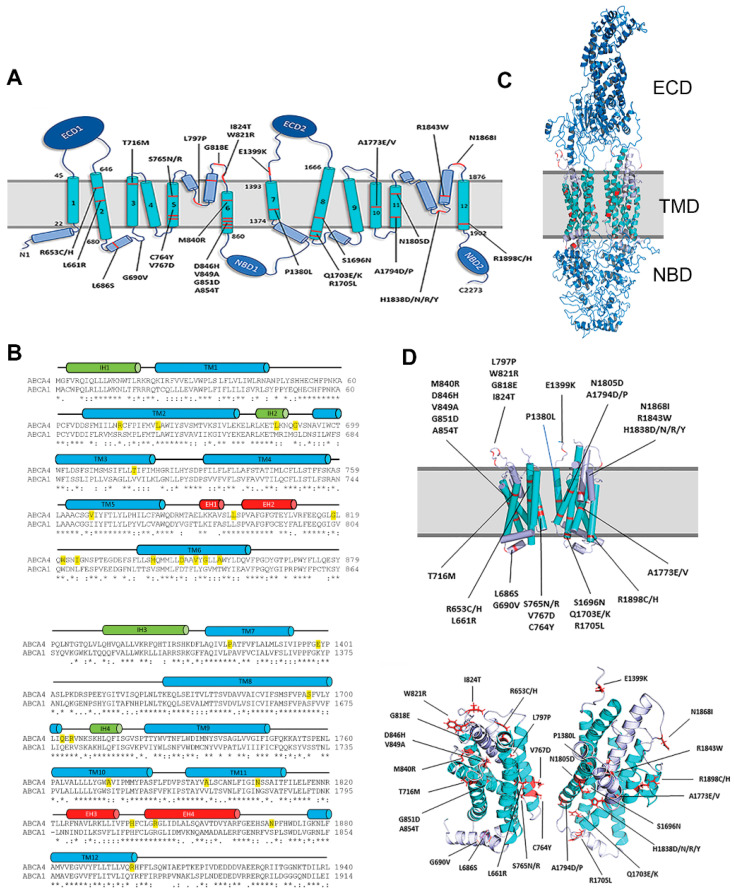
Location of TMD missense mutations in ABCA4 associated with STGD1. (**A**) Topological model of ABCA4 showing the location of mutations in TMD1 and TMD2. ECD—Exocytoplasmic Domain; TMD—Transmembrane Domain; NBD—Nucleotide Binding Domain. (**B**) Sequence alignment of the Transmembrane Domains TMD1 and TMD2 of ABCA4 and ABCA1 using Clustal Omega. The locations of the α-helical membrane spanning segments are shown as blue cylinders (TM1-12), intracellular transverse coupling helices are in green (IH1–IH4), and exoplasmic V-shaped α-helical hairpin helices are in red (EH1–EH4). The STGD1 missense variants examined in this study are highlighted in yellow. (**C**) Homology model of ABCA4 based on the cryoEM structure of ABCA1. (**D**) The location of disease-causing mutations within the TMD1 and TMD2. Top: Transverse view relative to the membrane with the transmembrane segments shown as cylinders. Bottom: Top view relative to the membrane with the transmembrane segments shown as ribbons.

**Figure 2 ijms-22-00185-f002:**
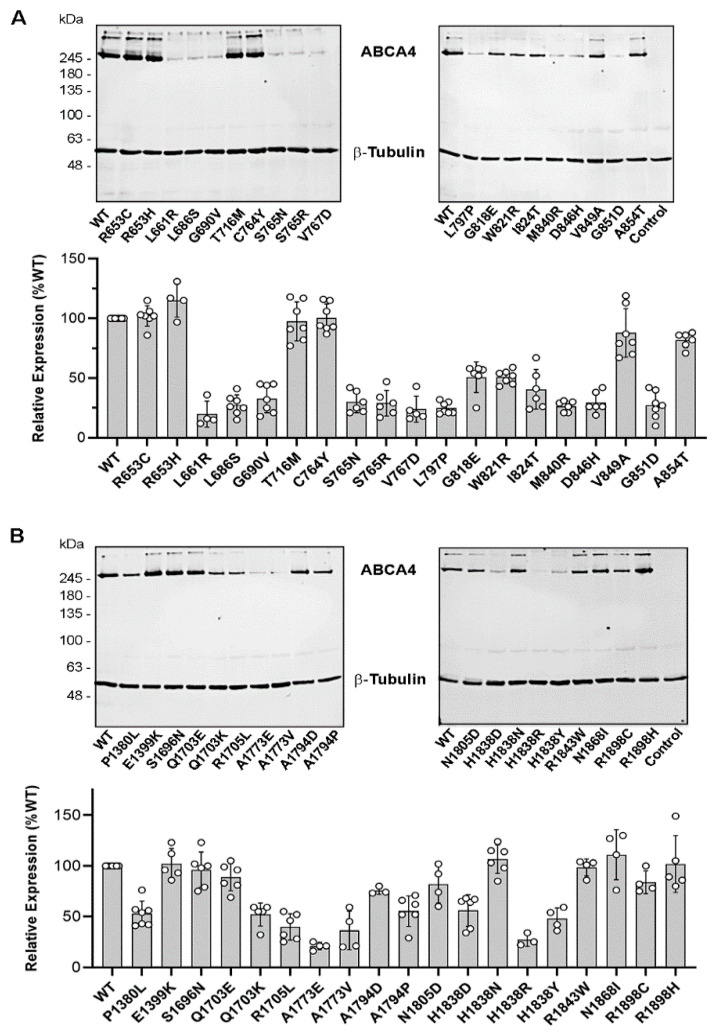
Solubilization of ABCA4 TMD variants. HEK293T cells expressing ABCA4 variants were solubilized in CHAPS. After the removal of aggregated material by centrifugation, the cell supernatants were resolved by SDS gel electrophoresis and analyzed on Western blots labeled for ABCA4 and β-tubulin as a loading control. (**A**) (Upper) Representative Western blots of TMD1 variants. (Lower) Quantification of TMD1 variants relative to WT ABCA4. (**B**) (Upper) Representative Western blots of TMD2 variants. (Lower) Quantification of TMD2 variants relative to WT ABCA4. The control sample represents non-transfected HEK293T cells. Quantified profiles are the mean of relative expression levels ± SD for *n* ≥ 4, where each point represents an independent experiment.

**Figure 3 ijms-22-00185-f003:**
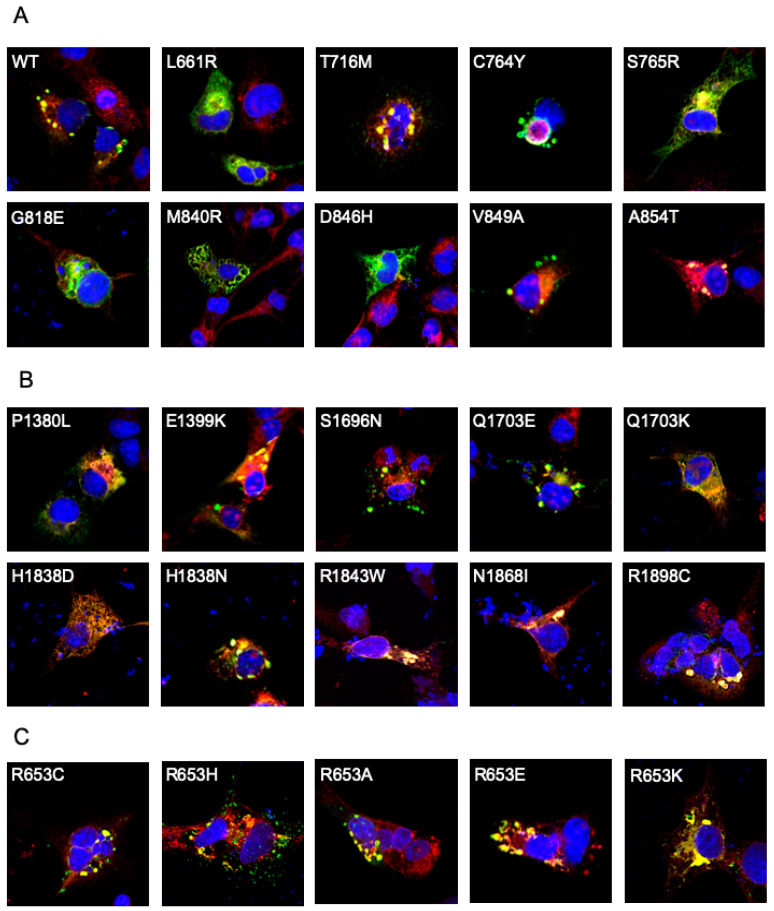
Cellular localization of TMD and R653 variants. Representative immunofluorescence micrographs of ABCA4 variants expressed in COS7 culture cells. The prevalence of vesicles is observed in cells expressing WT ABCA4 and variants with high levels of expression following CHAPS solubilization observed in Figure 2. Prevalence of reticular localization is observed in TMD variants with low solubilization. (**A**) Representative TMD1 variants; (**B**) Representative TMD2 Variants. (**C**) p.R653 Variants. GREEN—Rho1D4 antibody used to label ABCA4 variants containing the 1D4 tag; RED—anti-calnexin used as an ER marker; BLUE—DAPI (nucleus).

**Figure 4 ijms-22-00185-f004:**
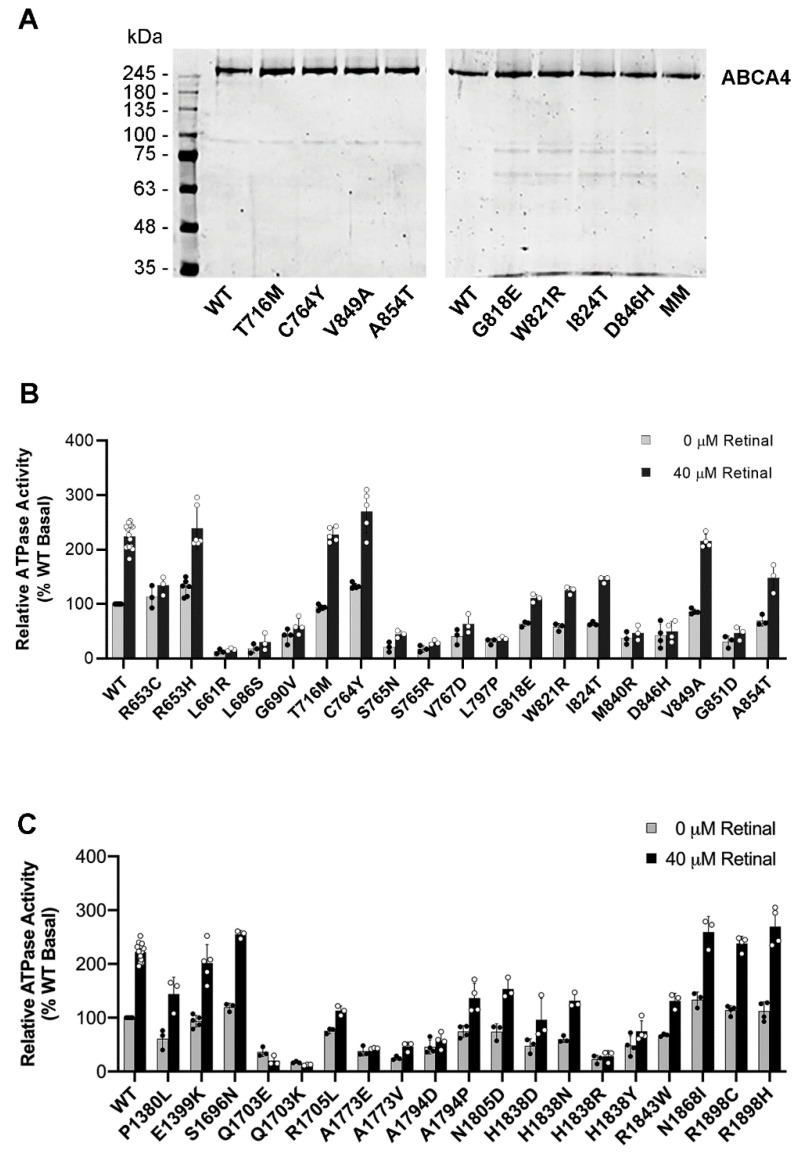
ATPase activity of ABCA4 TMD variants. TMD variants purified by immunoaffinity chromatography were analyzed for purity and protein concentration on SDS gels stained with Coomassie blue. (**A**) Representative SDS gels of purified variants stained with Coomassie Blue. (**B**) ATPase activity of the purified TMD1 variants in the absence and presence of 40 µM all-trans retinal; (**C**) ATPase activity of the TMD2 variants. Data show the mean activity ± SD for TMD1 and TMD2 variants *n* ≥ 3.

**Figure 5 ijms-22-00185-f005:**
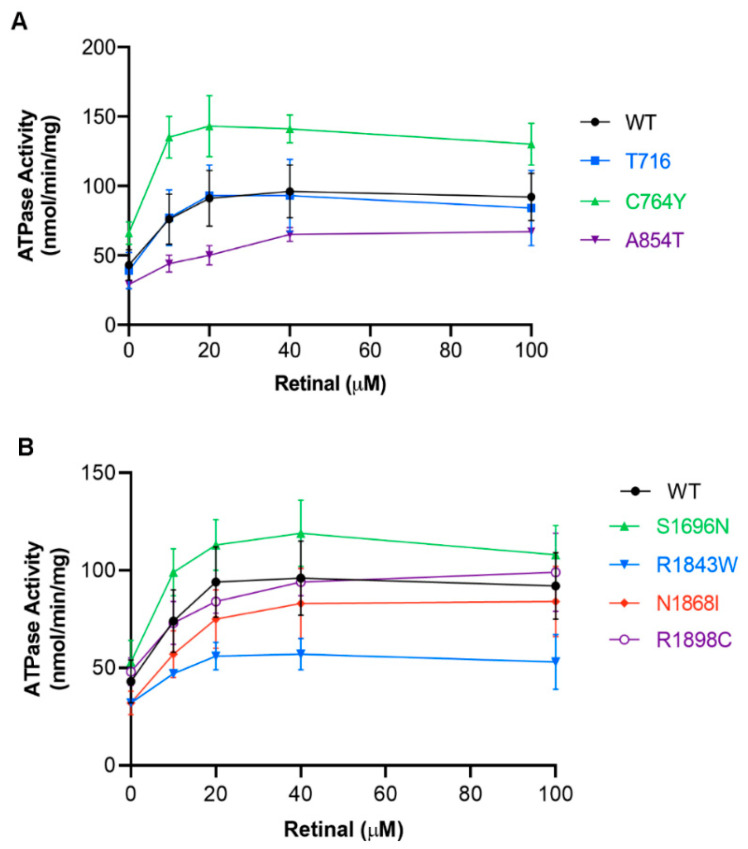
ATPase activity of selected TMD variants as a function of all-trans retinal concentration (Retinal). (**A**) Profiles of TMD1 variants. (**B**) Profiles of TMD2 variants. Specific ATPase activity (nmol of ATP hydrolyzed per minute per milligram of ABCA4) was calculated for the TMD variants. Experiments were performed at concentrations similar to WT ABCA4 and show the mean activity ± SD for *n* ≥ 2.

**Figure 6 ijms-22-00185-f006:**
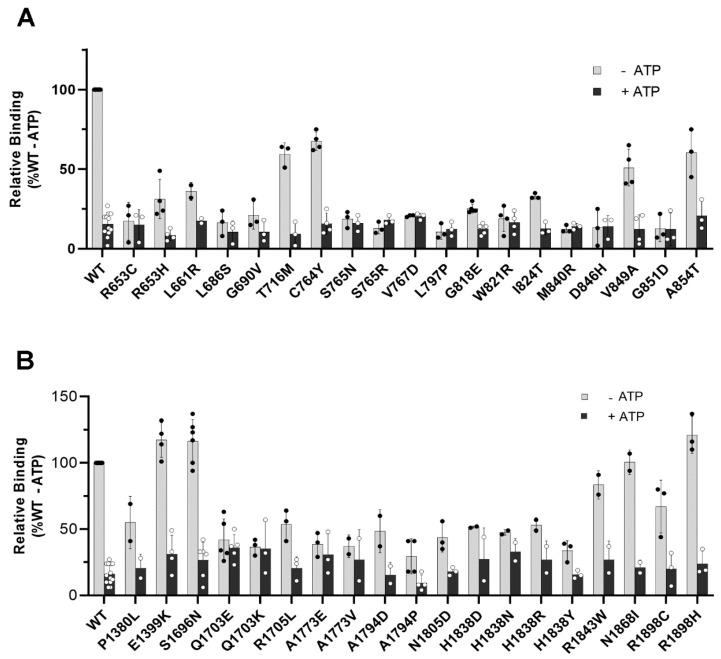
Binding of *N*-Ret-PE to TMD variants. ABCA4 TMD variants immobilized on an immunoaffinity matrix and containing bound *N*-Ret-PE were treated in the absence (grey) or presence (black) of 1 mM ATP. (**A**) *N*-Ret-PE binding profiles for TMD1 variants shown as a mean ± SD for *n* ≥ 3 and (**B**) TMD2 variants shown as a mean ± SD for *n* ≥ 2.

**Figure 7 ijms-22-00185-f007:**
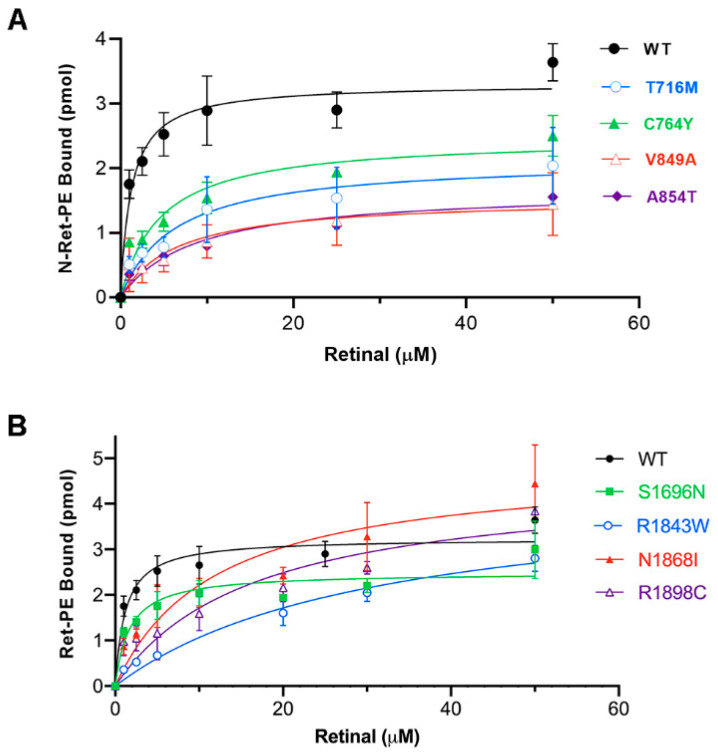
Binding of *N*-Ret-PE as a function of all-trans retinal concentration for selected TMD variants. ABCA4 TMD variants immobilized on an immunoaffinity matrix were treated with various concentrations of radiolabeled all-trans retinal. The amount of bound *N*-Ret-PE was determined after the removal of excess retinal. (**A**) Best fit binding curves for TMD1 variants (*n* ≥ 2) for the following apparent mean Kd ± SD; WT Kd = 1.2 ± 0.2 µM; p.T716M Kd = 6.3 ± 0.7 µM; p.C764Y Kd = 4.6 ± 0.4 µM; p.V849A Kd = 6.4 ± 0.9 µM; p.A854T Kd = 8.6 ± 0.9 µM. (**B**) Best fit binding curves for TMD2 variants for *n* ≥ 2; Kd = p.E1399K Kd = 2.8 ± 0.3 µM; p.S1696N Kd = 3.5 ± 0.3 µM; p.R1843W Kd = 31 ± 3.5 µM; p.N1868I Kd = 12.0 ± 0.8 µM; p.R1898C Kd = 16.0 ± 1.7 µM.

**Figure 8 ijms-22-00185-f008:**
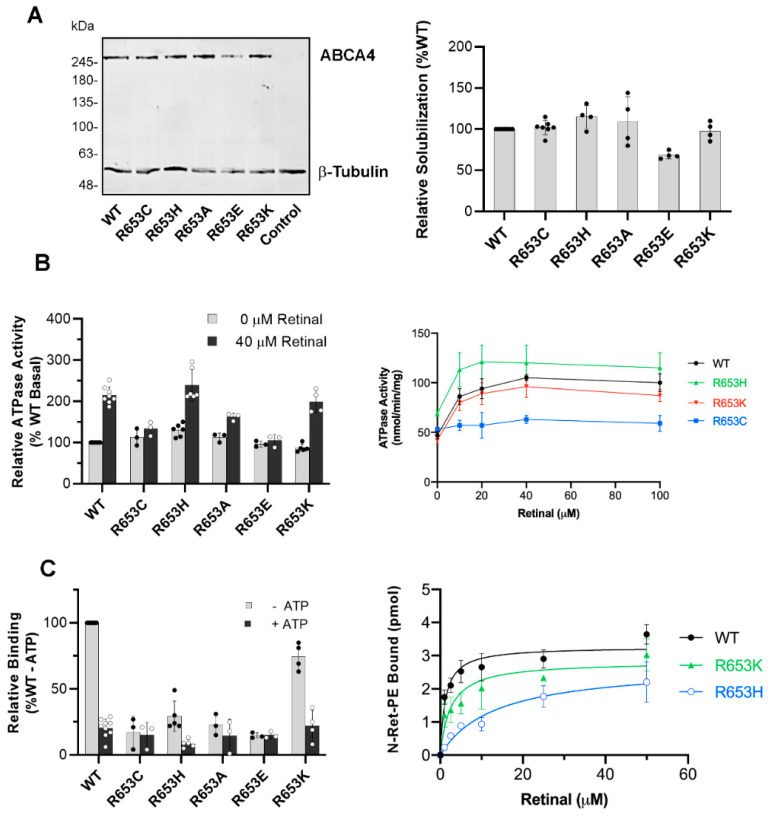
Biochemical analysis of R653 Variants. (**A**) (Left) Representative Western blots showing the CHAPS solubilization level of the p.R653 variants. (Right) Bar plot indicates the average relative expression of p.R653 variants ± SD (*n* ≥ 2). (**B**) (Left) Relative ATPase activity of p.R653 variants in the absence (grey) or presence (black) of 40 µM all-trans retinal (*n* ≥ 3). *p*-values with and without retinal: p.R653C (*p* = 0.021); p.R653H (*p* = 0.0005); p.R653A (*p* = 0.0005); p.R653E (*p*-value = 0.426); p.R653K (*p* = 0.0013). (Right) The specific AT ≥ ase activity was measured for p.R653C/H/K variants as a function of all-trans retinal concentration (*n* ≥ 2). (**C**) (Left) Relative binding to *N*-Ret-PE in the absence (grey) or presence (black) of 1 mM ATP. *p*-values with and without ATP: p.R653C *p* value = 0.422; p.R653H 0.0087; p.R653A *p* = 0.289; p.R653E *p* = 0.833; p.R653K *p* = 0.0043. (Right) Best fit retinal binding curves for the variants (*n* ≥ 2) with the following apparent mean Kd ± SD; WT Kd = 1.2 ± 0.2 µM; p.R653K Kd = 2.5 ± 1 µM and p.R653K Kd = 13.3 ± 2.5 µM.

**Figure 9 ijms-22-00185-f009:**
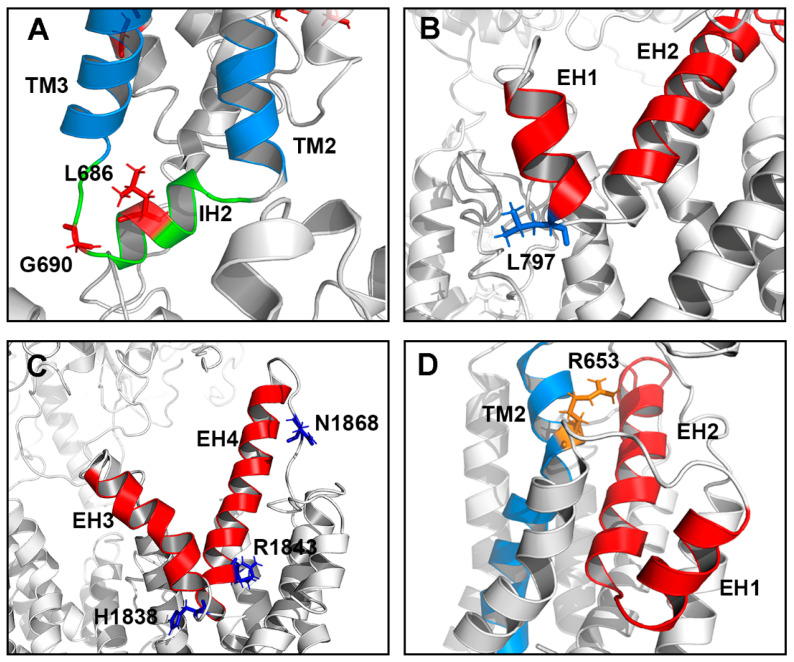
Location of selected amino acid residues in the ABCA4 I-Tasser homology model that, when mutated, cause STGD1. (**A**) The loop region between TM2 and TM3 (blue) showing the intracellular transverse helix IH2 (green) with residues L686 and G690. ABCA4 variants shown as sticks. p.L686S and p.G690V variants cause severe protein misfolding and are predicted to cause severe STGD1 in trans with a null ABCA4 allele. (**B**) The EH1 and EH2 helices (red) between TM5 and TM6 within TMD1 and showing the location of the L797 residue. The ABCA4 variant p.L797P causes significant protein misfolding and is predicted to cause severe STGD1 in trans with a null ABCA4 allele. (**C**) The EH3 and EH4 helices (red) between TM11 and TM12 within TMD2 and showing the H1838, R1843, and N1868 residues. ABCA4 variants p.H1838D, p.H1838N, p.H1838R, and p.H1838Y are predicted to cause moderate or severe STGD1 (see Table 2) in trans with a null ABCA4 allele. The p.R1843W variant is predicted to cause moderate STGD1 in trans with a null ABCA4 allele. The p.N1868I variant in the loop between EH4 and TM12 is predicted to cause a mild STGD1 phenotype based on functional assays. (**D**) Location of R653 within TM2. This residue is suggested to comprise part of the binding site for *N*-Ret-PE substrate based on biochemical studies. TM2 (blue); EH1 and EH2 (red).

**Table 1 ijms-22-00185-t001:** Summary of biochemical analysis of TMD1 variants.

Variant	Localization	Relative Solubilization	Basal ATPase Activity	*N*-Ret-PE-Induced ATPase Activity	*N*-ret-PE Binding no ATP	*N*-Ret-PE Binding with ATP	Class *	Predicted Severity
WT	Vesicles/ER	100	100	222 ± 22	100	15 ± 7	--	Normal
R653C	Vesicles/ER	102 ± 9	113 ± 21	134 ± 17	17 ± 12	15 ± 10	1	Severe
R653H	Vesicles/ER	115 ± 14	130 ± 15	239 ± 39	31 ± 12	9 ± 3	2	Moderate/ Mild *
L661R	ER	20 ± 11	13 ± 4	16 ± 3	36 ± 6	18 ± 2	1	Severe
L686S	ER	27 ± 8	18 ± 9	30 ± 16	16 ± 8	11 ± 7	1	Severe
G690V	ER	33 ± 12	41 ± 12	59 ± 14	21 ± 9	11 ± 7	1	Severe
T716M	Vesicles/ER	98 ± 16	94 ± 4	228 ± 13	59 ± 7	9 ± 8	3	Mild
C764Y	Vesicles/ER	100 ± 12	133 ± 5	270 ± 39	68 ± 6	16 ± 7	3	Mild
S765N	ER	30 ± 9	21 ± 9	45 ± 7	19 ± 5	16 ± 5	1	Severe
S765R	ER	29 ± 11	18 ± 6	29 ± 4	13 ± 4	18 ± 3	1	Severe
V767D	ER	24 ± 11	40 ± 14	63 ± 17	21 ± 1	20 ± 2	1	Severe
L797P	ER	25 ± 5	31 ± 5	37 ± 2	11 ± 5	12 ± 5	1	Severe
G818E	ER	51 ± 13	63 ± 5	111 ± 8	25 ± 3	12 ± 3	2	Moderate /Severe
W821R	ER	51 ± 6	58 ± 7	125 ± 7	19 ± 8	17 ± 6	2	Moderate /Severe
I824T	ER	41 ± 17	64 ± 3	145 ± 6	33 ± 2	13 ± 4	2	Moderate /Severe
M840R	ER	26 ± 4	37 ± 12	47 ± 14	12 ± 2	14 ± 2	1	Severe
D846H	ER	30 ± 9	42 ± 22	50 ± 18	13 ± 12	14 ± 7	1	Severe
V849A	Vesicles/ER	88 ± 20	86 ± 5	216 ± 13	51 ± 12	12 ± 9	3	Mild
G851D	ER	27 ± 11	31 ± 11	46 ± 12	13 ± 8	12 ± 10	1	Severe
A854T	Vesicles/ER	82 ± 6	70 +/−10	148 ± 26	60 ± 5	21 ± 9	2	Moderate

Values % WT ABCA4; * Placed in Class 2 (Moderate/Mild) due to low *N*-Ret-PE binding; Data: mean ± SD. * Classification is based equally on the relative expression and substrate-activated ATPase. For more details, see reference [19].

**Table 2 ijms-22-00185-t002:** Summary of biochemical analysis of TMD2 variants.

Variant	Localization	Relative Solubilization	Basal ATPase Activity	*N*-Ret-PE-Induced ATPase Activity	*N*-Ret-PE Binding no ATP	*N*-Ret-PE Binding with ATP	Class	Predicted Severity
WT	Vesicles/ER	100	100	222 ± 16	100	16 ± 7	--	Normal
p.P1380L	ER	53 ± 12	61 ± 19	144 ± 31	55 ± 20	21 ± 11	2	Moderate
p.E1399K	Vesicles/ER	102 ± 15	94 ± 11	202 ± 34	117 ± 13	31 ± 14	3	Mild
p.S1696N	Vesicles/ER	96 ± 17	120 ± 8	255 ± 7	116 ± 16	27 ± 13	3	Mild
p.Q1703E	Vesicles/ER	89 ± 13	36 ± 9	20 ± 9	42 ± 16	36 ± 10	1	Severe
p.Q1703K	ER	52 ± 11	17 ± 2	12 ± 1	37 ± 6	35 ± 20	1	Severe
p.R1705L	ER	40 ± 13	76 ± 5	113 ± 9	54 ± 12	21 ± 9	2	Moderate /Severe
p.A1773E	ER	21 ± 4	38 ± 9	43 ± 1	39 ± 9	31 ± 16	1	Severe
p.A1773V	ER	36 ± 20	25 ± 4	46 ± 9	37 ± 8	27 ± 23	1	Severe
p.A1794D	ER	76 ± 4	46 ± 13	57 ± 14	49 ± 16	16 ± 9	1	Severe
p.A1794P	ER	55 ± 15	74 ± 11	137 ± 27	30 ± 13	10 ± 6	2	Moderate /Severe
p.N1805D	Vesicles/ER	82 ± 18	74 ± 15	154 ± 19	44 ± 11	18 ± 3	2	Moderate
p.H1838D	ER	56 ± 16	48 ± 13	96 ± 40	52 ± 1	28 ± 23	2	Moderate /Severe
p.H1838N	Vesicles/ER	107 ± 14	60 ± 6	132 ± 13	48 ± 2	33 ± 10	2	Moderate
p.H1838R	ER	27 ± 6	23 ± 8	28 ± 11	53 ± 6	27 ± 14	1	Severe
p.H1838Y	ER	48 ± 11	48 ± 18	75 ± 20	34 ± 8	16 ± 3	1	Severe
p.R1843W	Vesicles/ER	99 ± 8	68 ± 3	131 ± 14	84 ± 11	27 ± 14	3	Moderate/Mild
p.N1868I	Vesicles/ER	111 ± 25	134 ± 14	259 ± 29	101 ± 9	21 ± 6	3	Mild
p.R1898C	Vesicles/ER	84 ± 11	114 ± 8	238 ± 13	67 ± 20	20 ± 13	3	Mild
p.R1898H	Vesicles/ER	102 ± 28	112 ± 17	270 ± 35	121 ± 14	24 ± 10	3	Mild

Values % WT ABCA4; Data: mean ± SD.

## Data Availability

The data that support the findings of this study are available from the corresponding author upon reasonable request.

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
