# Peer review of "Functional Characterization of ABCA4 Missense Variants Linked to Stargardt Macular Degeneration"

_ijms, 2020, doi:10.3390/ijms22010185_

Round 1

Reviewer 1 Report

The study by Garces et al. describes functional analyses of ABCA4 disease-causing variants (mostly) in the transmembrane domains of ABCA4 and classifies patients into three groups based on disease severity (age of onset) and according to the impact of the analyzed variants on the protein function based on the protein expression, ATP binding, and ATPase activity analyses with and without the native ABCA4 substrate, N-Ret-PE. The study suggests functional grouping of ABCA4 missense alleles into three classes, severe, moderate and mild, based on the cumulative data from all functional analyses.

This study continues functional analyses of ABCA4 from the same laboratory for many years, now already decades, which have resulted in consistent improvement of our understanding of retinal pathology caused by ABCA4 mutations. The methods, results, analysis and discussion are very well performed, and I do not have really any substantial, or even marginal, criticism to add.

A couple of minor comments:

  1. The genetic nomenclature could be changed to the currently accepted version throughout the manuscript, i.e., to 3-letter format for amino acids. For example, the R653C could/should read p.Arg653Cys, etc.
  2. Stargardt disease is officially called “Stargardt macular dystrophy”, but there is no crime to use “degeneration”. Some clinicians tend to call upon the difference between “dystrophy” and “degeneration”.
  3. The authors describe how they got to the classification of the three groups in Table 1 and in the text. While maybe a simplification, it is sufficient for the current study setting. Is there a specific weight of any of the 6 analyses (expression, localization, ATP binding, etc.) to override others, if these do not conform uniformly to the grouping? Is the ATPase assay the deciding analysis of the six? It might be beneficial to add a sentence or two to the Table 1 legend about it. How many exceptions are there in the entire set of analyzed mutations?

In summary, it is an important study which will add to our knowledge of the disease-causing ABCA4 variation.

Author Response

1.    The genetic nomenclature could be changed to the currently accepted version throughout the manuscript, i.e., to 3-letter format for amino acids. For example, the R653C could/should read p.Arg653Cys, etc.

Response:  We have included p. in front of the Variants, but retained the single letter for the amino acid residue as permitted by the journal.  For further clarification we have added the 1 letter format to the Abbreviation section to identify the single letter amino acid.

2.   Stargardt disease is officially called “Stargardt macular dystrophy”, but there is no crime to use “degeneration”. Some clinicians tend to call upon the difference between “dystrophy” and “degeneration”

Response:  We are aware that some clinicians prefer macular dystrophy for inherited macular degeneration.  However, many researchers including us prefer the use macular degeneration. Therefore, since either can be used we have retained macular degeneration.

3.    The authors describe how they got to the classification of the three groups in Table 1 and in the text. While maybe a simplification, it is sufficient for the current study setting. Is there a specific weight of any of the 6 analyses (expression, localization, ATP binding, etc.) to override others, if these do not conform uniformly to the grouping? Is the ATPase assay the deciding analysis of the six? It might be beneficial to add a sentence or two to the Table 1 legend about it. How many exceptions are there in the entire set of analyzed mutations?

Response:  We thank the reviewer for his/her comment/question.  The classifications are primarily based on equal weighting of the expression and substrate-activated ATPase activity assays as first formulated in an early reference (see reference 19).  We have now added this as a legend to Table 1. [page 6; line 139-140]   In this data set we have not observed any major exceptions.  However, as indicated in the manuscript the degree of severity of STGD1 including the age of onset has not been thoroughly reported for all variants, so this awaits further analysis.

Reviewer 2 Report

Garces et al investigates 38 mutations in ABCA4 specifically to determine the properties of the missense causing variants. Subdivision of classes due to expression and functional properties was performed, to discuss the molecular properties that would contribute to phenotypic variation. The methodology is very well done and thorough. I have some minor comments but this paper would be a wonderful contribution to the scientific literature.

Minor comments:

1) How were all the tools combined to definitively identify the membrane spanning segments? Could this methodology be elaborated on? Ie, pooling is mentioned in methods, but how did that occur? Overlapping predictions using which methodology?
Models (page 20, line 9) should not be in bold.

2) Was any verification for the Rho 1D4 antibody performed before using, since it was created in-house?
3) On page 12, it is indicated that the addition of ATP induced a conformational change- is this the only way that further release of the substrate can occur?

4) The phenotypic discussion regarding a variant and mild phenotype- how many samples were examined to identify the P1380L variant as being not of Class 2 but of Class 3 when age was increased?

Author Response

1) How were all the tools combined to definitively identify the membrane spanning segments? Could this methodology be elaborated on? Ie, pooling is mentioned in methods, but how did that occur? Overlapping predictions using which methodology?
Models (page 20, line 9) should not be in bold.

Response:  The membrane spanning segments predicted from each algorithm were compared; the overlapping segments predicted in the majority of the algorithms were used to predict the membrane spanning segments.  These were then compared with those identified in ABCA1.  We have now elaborated on this in the methods section. [page 19; line 484-485].
The bold format of model has been removed.

2. Was any verification for the Rho 1D4 antibody performed before using, since it was created in-house?

Response:  The Rho 1D4 antibody was generated in house in 1982 and has now been used and verified in hundreds of applications world-wide for detection of 1D4 engineered sequences. It is available through Millipore-Sigma. This information has been added to the manuscript in the Methods section. [page 20; line 500-501]

3) On page 12, it is indicated that the addition of ATP induced a conformational change- is this the only way that further release of the substrate can occur?

Response: The substrate binds to ABCA4 with high affinity in the absence of ATP and hence previous efforts to dissociate the substrate by reduction in substrate concentration did not release the substrate.  The only known nondenaturating way to remove substrate is by the addition of ATP or related nonhydrolyzable ATP analogs which are known to cause a conformational change in ABCA4 dissolving the high affinity binding site and thus releasing (transporting) the substrate.

4) The phenotypic discussion regarding a variant and mild phenotype- how many samples were examined to identify the P1380L variant as being not of Class 2 but of Class 3 when age was increased?

Response: The P1380L is a class 2 variant (moderate); our search of the literature indicated that individuals homozygous for the P1308L variant (total number of five) have an age of onset and severity within the moderate range.  5 homozygous individuals were found in our literature search.